# Bridging the Gap between Database Search and *De Novo* Peptide Sequencing with SearchNovo

**Jun Xia**[1,2†,*] **Sizhe Liu**[3†], **Jingbo Zhou**[1,2†], **Shaorong Chen**[1,2], **Hongxin Xiang**[5],
**Zicheng Liu**[1,2], **Yue Liu**[4], **Stan Z. Li**[1*]

[1]Westlake University, [2]Zhejiang University, [3] University of Southern California,
[4] National University of Singapore, [5]Hunan University
{xiajun, zhoujingbo, Stan.ZQ.Li}@westlake.edu.cn; sliu07270@usc.edu

## ABSTRACT

Accurate protein identification from mass spectrometry (MS) data is fundamental to unraveling the complex roles of proteins in biological systems, with peptide sequencing being a pivotal step in this process. The two main paradigms for peptide sequencing are database search, which matches experimental spectra with peptide sequences from databases, and *de novo* sequencing, which infers peptide sequences directly from MS without relying on pre-constructed database. Although database search methods are highly accurate, they are limited by their inability to identify novel, modified, or mutated peptides absent from the database. In contrast, *de novo* sequencing is adept at discovering novel peptides but often struggles with missing peaks issue, further leading to lower precision. We introduce SearchNovo, a novel framework that synergistically integrates the strengths of database search and *de novo* sequencing to enhance peptide sequencing. SearchNovo employs an efficient search mechanism to retrieve the most similar peptide spectrum match (PSM) from a database for each query spectrum, followed by a fusion module that utilizes the reference peptide sequence to guide the generation of the target sequence. Furthermore, we observed that dissimilar (noisy) reference peptides negatively affect model performance. To mitigate this, we constructed pseudo reference PSMs to minimize their impact. Comprehensive evaluations on multiple datasets reveal that SearchNovo significantly outperforms state-of-the-art models. Also, analysis indicates that many retrieved spectra contain missing peaks absent in the query spectra, and the retrieved reference peptides often share common fragments with the target peptides. These are key elements in the recipe for SearchNovo's success. The code is available at: https://github.com/junxia97/SearchNovo.

## 1 INTRODUCTION

The identification of the proteins present in collected biological samples is a fundamental task in biomedicine, steering a better understanding of disease etiology and pathology, which is essential for the identification of new therapeutic targets for developing new treatments or drugs (Uzozie & Aebersold, 2018; Lin et al., 2020). Tandem mass spectrometry (MS/MS) stands out as the only high-throughput technique capable of analyzing the protein composition in biological samples due to its high sensitivity and specificity (Aebersold & Mann, 2003). In bottom-up proteomics (Zhang et al., 2013), proteins are digested into smaller peptide fragments, which are then analyzed using mass spectrometry to determine the amino acid sequences and, ultimately, to identify the original proteins.

The core of protein identification lies in the challenge of peptide sequencing, where the goal is to determine the peptide amino acid sequence for each observed mass spectrum. Two primary paradigms exist to tackle this problem: database search (Nesvizhskii, 2010; Griss, 2016) and *de novo* peptide sequencing (Tran et al., 2017). Database search approaches involve matching the observed spectra to pre-constructed peptide-spectrum match (PSM) databases, selecting the most similar match as the identification results. For example, SEQUEST (Eng et al., 1994) matches observed MS/MS against a

---

*†Equal Contribution, *Corresponding Author

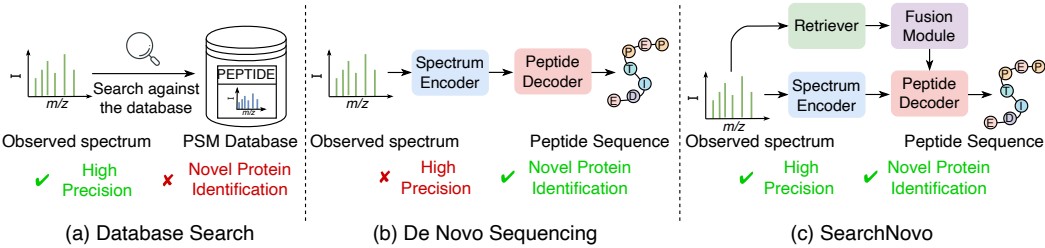

Figure 1: The semantic diagrams of database search, *de novo* sequencing and our SearchNovo.

protein sequence database and scores the matches based on the correlation between experimental and peptides' theoretical spectra. MaxQuant (Cox & Mann, 2008) uses the Andromeda search engine for database search, allowing for highly accurate protein identification, especially in large-scale proteomics experiments. Despite the high precision, these tools are inherently limited by the scope of the database and cannot identify novel proteins that are not included in the pre-constructed database.

On the other hand, *de novo* sequencing methods predict peptide sequences directly from the observed mass spectra without dependence on external databases. This makes it indispensable for applications where databases are incomplete or unavailable, such as antibody sequencing (Tran et al., 2016), human leukocyte antigen (HLA) neoantigen discovery (Tran et al., 2020), and the identification of novel proteins and peptides not yet cataloged in existing databases (Vitorino et al., 2020). This family began with early rule-based methods that manually interpreted tandem mass spectra and infer peptide sequences (Eng et al., 1994; Li et al., 2005; Kong et al., 2017). Modern deep learning methods, such as DeepNovo (Tran et al., 2017), PointNovo (Qiao et al., 2021), InstaNovo (Eloff et al., 2023) and Casanovo (Yilmaz et al., 2022), usually train encoder-decoder models to predict peptide sequence for observed spectrum, which have improved the models' ability to identify novel proteins. Despite the remarkable success, the performance of these methods remains unsatisfactory, partly due to missing signal peaks in mass spectra data and a lack of additional cues to guide peptide sequence inference.

To capitalize on the strengths and compensate for the weaknesses of the above two primary paradigms, we integrate database search and *de novo* sequencing into a unified framework, SearchNovo, which contains two core modules: retriever and fusion module. In the retriever, for each query spectrum, we design a novel and efficient strategy to search for the top one most similar spectrum from the database with mass constraints. In fusion module, we employ a novel fusion layer to fuse the prefix peptide sequence and retrieved peptide sequence, with the expectation that the latter can provide the clues for the generation of the target peptide sequence. Additionally, when numerous query spectra are matched with low-similarity reference peptide sequences (noisy reference peptides) from the database, the model's performance degrades significantly. To further address this issue, we construct pseudo reference PSMs that could prevent the model from over-relying on noisy reference spectra when generating target peptide sequences. We highlight the core contributions of this work as follows:

- As illustrated in Figure 1, we integrate two primary paradigms—database search and *de novo* sequencing—into a unified framework called SearchNovo, enjoying the strengths of both paradigms: high sequencing precision and the strong ability to identify novel proteins.

- We design the retriever and fusion module to maximize the utilization of the retrieved reference PSMs to guide the generation of target peptide sequence. Additionally, we implemented a straightforward yet effective strategy to reduce the risk of the model over-relying on noisy (dissimilar) reference spectra when inferring the target peptide sequence.

- As revealed in section 5.4, the retrieved PSMs by SearchNovo contain the missing signal peaks in the query spectra and common peptide fragments in the target sequence, resulting in its superior performance compared to state-of-the-art methods across multiple datasets.

## 2 BACKGROUND

Protein identification is a critical procedure in discovering drug targets and disease biomarkers, addressing a major bottleneck in the AI for Drug Discovery and Development (AIDD) pipeline. While the AI community has focused extensively on drug design with known protein targets (Xia

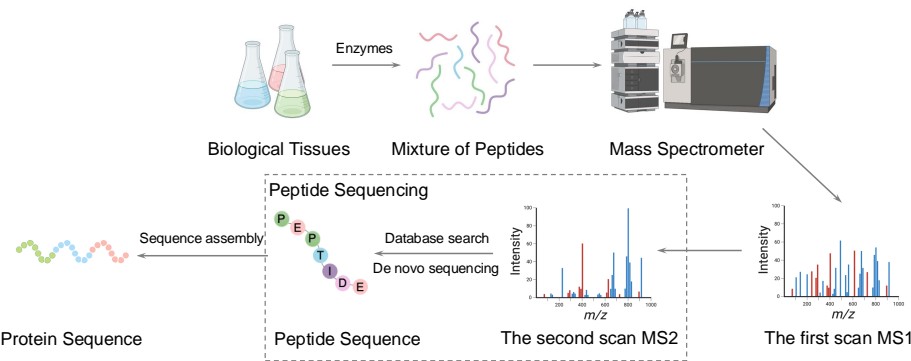

Figure 2: The semantic diagram of protein identification using mass spectrum.

et al., 2023b;a), the challenge of identifying these key protein targets using AI remains underexplored. In this section, we provide a brief overview of the general pipeline for protein identification using mass spectrometry, aiming to foster greater understanding and attention to this important task from the AI community.

As depicted in Figure 2, the process of mass spectrometry-based proteomics typically begins with the digestion of proteins into smaller peptides. These peptides are then ionized and introduced into the mass spectrometer for analysis. Identifying peptide sequences primarily involves two crucial steps: the first scan (MS1) and the second scan (MS2). During the MS1 phase, the mass spectrometer measures the mass-to-charge ratio ($m/z$) of intact peptide ions, generating a spectrum that displays peaks corresponding to various peptides, referred to as precursors. Each peak signifies a specific peptide present in the biological sample, and its intensity indicates the peptide's relative abundance. In the subsequent MS2 phase, a chosen precursor ion undergoes fragmentation into smaller ions, typically at the peptide bond level, yielding a detailed spectrum known as the MS2 spectrum. Each peak in this spectrum comprises a tuple that includes the $m/z$ value and its associated intensity. The goal of peptide sequencing is to deduce the amino acid sequence of the peptide directly from the MS2 spectra and the precursor information (mass and charge of the intact peptide). Ultimately, we can reconstruct the entire protein sequence using protein sequence assembly techniques (Liu et al., 2015).

## 3    RELATED WORK

There exist two lines of works for peptide sequencing. The first line is database search, where we compare the observed mass spectra against the theoretical fragmentation mass spectra of peptide sequences in the database and pick the peptide sequence with the highest matching score as the identification result. Typical methods and tools include SEQUEST (Eng et al., 1994), pFind (Li et al., 2005), MaxQuant (Cox & Mann, 2008), MSFragger (Kong et al., 2017) and Open-pFind (Sun et al., 2019). However, these methods cannot sequence the peptides out of the pre-constructed database. The second line of works is *de novo* peptide sequencing, where we predict the peptide sequences for observed spectra without relying on pre-constructed databases. Initially, researchers cast the *de novo* peptide sequencing task as finding the largest path in the spectrum graph (Dančík et al., 1999; Taylor & Johnson, 2001) or compute the best sequences whose fragment ions can best interpret the peaks in the observed MS2 spectrum using Hidden Markov Model (Fischer et al., 2005) or dynamic programming algorithm (Ma et al., 2003). With the prosperity of deep learning, DeepNovo (Tran et al., 2017) is the first method applying deep neural networks to the task of *de novo* peptide sequencing. It regards the task as the image caption (Stefanini et al., 2022) in computer vision and incorporates the encoder-decoder architecture to predict the peptide sequence. To annotate the high-resolution MS data, PointNovo (Qiao et al., 2021) adopts an order invariant network structure for peptide sequencing. More recently, Casanovo (Yilmaz et al., 2022) first employs a transformer encoder-decoder architecture (Vaswani et al., 2017) to predict the peptide sequence for the observed spectra. Following Casanovo, AdaNovo (Xia et al., 2024) proposes conditional mutual information-based re-weighting methods to help identify Post-Translational Modifications (PTMs). Despite the remarkable advancements, the performance of *de novo* peptide sequencing methods remains inferior

to database search, partially due to missing signal peaks in mass spectrum data and a lack of additional clues to guide peptide sequence generation. We recommend readers refer to a recent survey (Xia et al., 2025) for more detailed information.

# 4 METHOD

## 4.1 FORMULATION

We represent the peaks in an MS2 spectrum as $\mathbf{x} = \{(m_i, t_i)\}_{i=1}^{M}$, where each peak is defined by a pair $(m_i, t_i)$, with $m_i$ representing the mass-to-charge ratio (*m/z*) and $t_i$ representing the intensity. The number of peaks, $M$, varies between spectra. The precursor ion is described as $\mathbf{z} = (m_{prec}, c_{prec})$, where $m_{prec} \in \mathbb{R}$ represents the precursor mass (the total mass of the peptide sequence to be predicted), and $c_{prec} \in \{1, 2, \ldots, 10\}$ is the charge state. A peptide sequence is denoted by $\mathbf{y} = (y_1, y_2, \ldots, y_N)$, where $y_i$ is the $i$-th amino acid in the sequence, and $N$ is the total number of amino acids in the peptide. The prefix subsequence of $\mathbf{y}$ up to position $j$ is written as $\mathbf{y}_{<j}$.

The task of *de novo* peptide sequencing models is to predict each amino acid $y_j$ conditioned on the MS2 spectrum $\mathbf{x}$, the precursor ion $\mathbf{z}$, and the previously generated sequence $\mathbf{y}_{<j}$. The probability distribution for a peptide sequence is modeled as:

$$P(\mathbf{y} \mid \mathbf{x}, \mathbf{z}; \theta) = \prod_{j=1}^{N} p(y_j \mid \mathbf{y}_{<j}, \mathbf{x}, \mathbf{z}; \theta), \tag{1}$$

where $j$ is the index for the current amino acid position, and $\theta$ represents the model parameters. Considering that both $\mathbf{x}$ and $\mathbf{z}$ can be derived from the spectrum, for simplicity, we will refer to them collectively as $\mathbf{x}$ in the following discussion. Common approaches, such as those in (Tran et al., 2017; Yilmaz et al., 2022; Xia et al., 2024), minimize the negative log-likelihood to optimize the model:

$$\ell(\theta) = -\sum_{j=1}^{N} \log p(y_j \mid \mathbf{y}_{<j}, \mathbf{x}; \theta). \tag{2}$$

During inference, the models typically use autoregressive decoding to predict each amino acid and apply heuristic search like beam search to generate candidate sequence $\mathbf{y}^*$.

In contrast, database search approaches solve the peptide sequencing problem by comparing the observed spectrum to a database of known peptides. For a given spectrum $\mathbf{x}$, the goal is to identify the peptide $\mathbf{y}^*$ in the database that best matches the spectrum by maximizing a similarity function:

$$\mathbf{x}^*, \mathbf{y}^* = \arg \max_{(\mathbf{x}', \mathbf{y}') \in \mathcal{D}} \text{sim}(\mathbf{x}, (\mathbf{x}', \mathbf{y}')), \tag{3}$$

where $\mathcal{D}$ is the PSM database. Various similarity scoring methods, including cross-correlation (Eng et al., 1994; 2013) or machine learning-based models (Liu et al., 2024; Degroeve & Martens, 2013), are used to compute $\text{sim}(\mathbf{x}, (\mathbf{x}', \mathbf{y}'))$, and the top-scoring peptides are returned as the predictions.

## 4.2 MODEL ARCHITECTURES

As shown in Figure 3, SearchNovo consists of a peak embedding layer (peak encoder), spectrum encoder, fusion layer and peptide decoder. In order to feed the mass spectrum peaks to the spectrum encoder, we regard each mass spectrum peak $(m_i, t_i)$ as a 'word' in natural language processing and obtain its embedding $h_i$ by individually encoding its *m/z* value ($m_i$) and intensity value ($t_i$) before combining them through summation. The detailed description of the peak embedding layer can be found in Appendix A.1. And then, we feed the peak embeddings $\mathbf{h} = \{h_i\}_{i=1}^{M}$ to the spectrum encoder that consists of multiple transformer layers with Multi-Head Self-Attention (MHA). Similar to natural language processing (Kenton & Toutanova, 2019), we select the top 150 peaks with the highest intensity values. If the number of peaks exceeds 150, only these top 150 peaks are used, as they are more likely to represent signal peaks. If the number of peaks is fewer than 150, we pad the sequence to 150 embeddings using a special token [PAD].

As for the peptide sequence, the amino acid vocabulary encompasses the 20 canonical amino acids, a special [EOP][1] token indicating the end of peptide sequence and several Post-Translational

---

[1] End Of Peptide

Modifications (PTMs). Here, the PTMs can be regarded as variants of canonical amino acids. We summarize the types of PTMs in the experimental datasets in Appendix A.3. And then, we employ an amino acid embedding layer (denated as AA. Embedding in Figure 3), a learnable lookup table that maps each token in the vocabulary to a fixed-size vector. Also, we apply the positional embedding (denoted as Pos. Embedding in Figure 3) from Transformer (Vaswani et al., 2017) to capture the positional information of each amino acid within the peptide sequence. The final amino acid embedding is obtained by summing AA. Embedding and Pos. Embedding. Similarly, we obtain the precursor embedding by individually encoding its mass ($m_{prec}$) and charge state ($c_{prec}$) before combining them through summation. The detailed embedding process is formulated in Appendix A.1. And then, we prepend the precursor embedding to the amino acid embedding sequence and input this combined sequence into the fusion layer. Kindly note that, for simplicity, we have omitted the precursor embeddings before all peptide sequences in Figure 3. Finally, we feed the fused embedding sequence $\mathbf{y}'_{<j}$ to the peptide encoder, which contains multiple identical transformer layers with causally masked multi-head self-attention and cross-attention to decode the target peptide sequence.

## 4.3 RETRIEVER

For each query spectrum, we aim to search for the most similar peptide-spectrum match (PSM) from the database. The most straightforward approach is to compute the similarity between the query spectrum and every spectrum in the database, selecting the peptide sequence corresponding to the spectrum with the highest similarity score as the reference. However, this method is computationally expensive. To mitigate this issue, we first narrow the database search range by leveraging the precursor mass of the query spectrum, which represents the total mass of the peptide sequence. Specifically, for each query spectrum $\mathbf{x}$, we select a subset of PSMs $\mathcal{D}_{\mathbf{x}}$ from the database $\mathcal{D}$ where the difference between the precursor mass and peptide

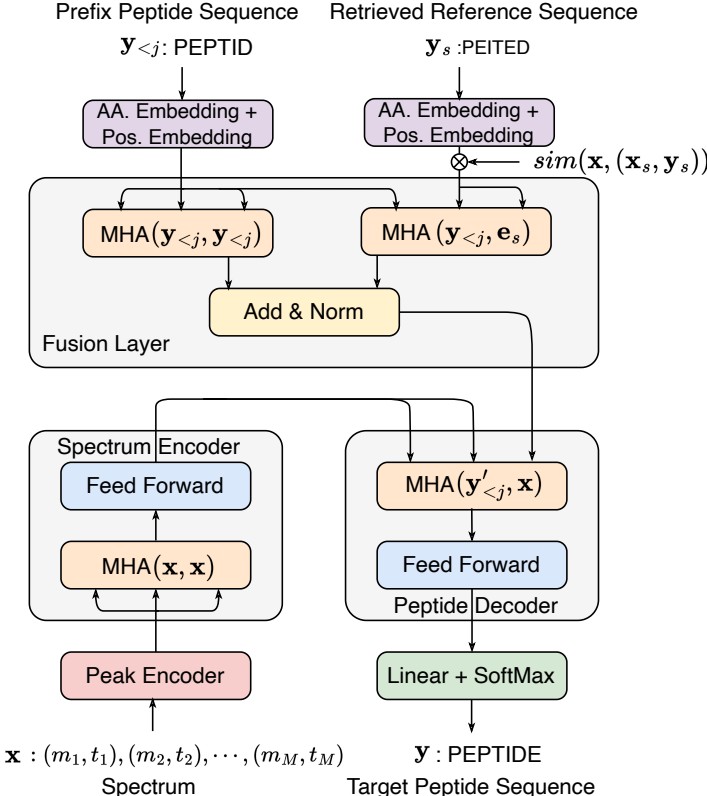

Figure 3: The overview of SearchNovo. We provide the formulation of MHA($\cdot, \cdot$) in Appendix A.2.

amino acids total mass is within $\pm 20$ Da (Dalton, a unit of mass commonly used to express the mass of atoms, molecules, and subatomic particles). If no candidate PSMs are found (i.e., $\mathcal{D}_{\mathbf{x}} = \emptyset$), we iteratively expand the mass tolerance window to $\pm 30$, $\pm 40$, $\pm 50$ Da, and so on, until at least one candidate PSM is identified ($\mathcal{D}_{\mathbf{x}} \neq \emptyset$). We then calculate the similarity between $\mathbf{x}$ and each spectrum in $\mathcal{D}_{\mathbf{x}}$ using the widely-used spectral similarity score MatchMS (Florian et al., 2020; de Jonge et al., 2024), selecting the peptide sequence $\mathbf{y}_s$ corresponding to the most similar spectrum $\mathbf{x}_s$ as the reference peptide sequence:

$$\mathbf{x}_s, \mathbf{y}_s = \arg \max_{(\mathbf{x}', \mathbf{y}') \in \mathcal{D}_{\mathbf{x}}} \text{sim}(\mathbf{x}, (\mathbf{x}', \mathbf{y}')). \tag{4}$$

Now, the *de novo* peptide sequencing task is to predict each amino acid $y_j$ with the spectrum $\mathbf{x}$, the precursor $\mathbf{z}$, the prefix peptide sequence $\mathbf{y}_{<j}$ and the reference peptide sequence $\mathbf{y}_s$. And thus, the

probability distribution of the peptide sequence $\mathbf{y}$ in Eq. 1 can be re-defined as,

$$P(\mathbf{y} \mid \mathbf{x}, \mathbf{x}_s, \mathbf{y}_s; \theta) = \prod_{j=1}^{N} p(y_j \mid \mathbf{y}_{<j}, \mathbf{x}, \mathbf{x}_s, \mathbf{y}_s; \theta), \tag{5}$$

where $\theta$ denotes the model parameters. In the experiments, we regard the training set as the database and search for the most similar PSM (excluding itself) for each query spectrum in the training set using the above retriever, and then we store these results for use in subsequent model training.

## 4.4 FUSION LAYER

The fusion layer contains two multi-head attention mechanisms (MHA). The first, denoted as $\mathrm{MHA}(\mathbf{y}_{<j}, \mathbf{y}_{<j})$, follows the standard Transformer architecture and operates over the prefix sequence $\mathbf{y}_{<j}$. The second, $\mathrm{MHA}(\mathbf{y}_{<j}, \mathbf{y}_s)$, is designed to extract information from the reference peptide sequence, where the query comes from $\mathbf{y}_{<j}$ and the key and value are derived from the representation of the reference sequence $\mathbf{y}_s$. Due to the limited space, we provide the detailed formula for $\mathrm{MHA}(\cdot, \cdot)$ in Appendix A.2. Considering that if a reference PSM $(\mathbf{x}_s, \mathbf{y}_s)$ is with high similarity to the query spectrum $\mathbf{x}$, $\mathbf{y}_s$ should be more helpful to infer the target peptide sequence. Therefore, we take the similarity into consideration when embed the peptide sequence. Formally,

$$\mathbf{e_s} = \mathrm{sim}(\mathbf{x}, (\mathbf{x}_s, \mathbf{y}_s)) \times \mathbf{E}_s, \tag{6}$$

where $\mathbf{E}_s$ is the peptide embedding from the AA. Embedding $E_{AA}(\cdot)$ and Pos. Embedding $E_{Pos.}(\cdot)$ layer,

$$\mathbf{E}_s = [E_{AA.}(y_1) + E_{Pos.}(y_1), \cdots, E_{AA.}(y_N) + E_{Pos.}(y_N)], \tag{7}$$

after applying these two parallel attention mechanisms, the outputs are combined using an Add & Norm (layer normalization) operation:

$$\mathbf{y}'_{<j} = \mathrm{Norm}\left(\mathrm{MHA}(\mathbf{y}_{<j}, \mathbf{y}_{<j}) + \mathrm{MHA}(\mathbf{y}_{<j}, \mathbf{e}_s)\right), \tag{8}$$

producing a new sequence $\mathbf{y}'_{<j}$, which is then used as the query input for the subsequent multi-head cross attention (i.e., $\mathrm{MHA}(\mathbf{y}'_{<j}, \mathbf{x})$). The following sub-layer is the same as standard Transformer model (Vaswani et al., 2017).

## 4.5 ROBUSTNESS TO THE NOISY REFERENCE PSMs

With the reference PSM $(\mathbf{x}_s, \mathbf{y}_s)$, we can redefine the training loss in Eq. 2 for SearchNovo as,

$$\ell(\theta) = -\sum_{j=1}^{N} \log p(y_j \mid \mathbf{y}_{<j}, \mathbf{x}, \mathbf{x}_s, \mathbf{y}_s; \theta). \tag{9}$$

However, as shown in Figure 5, we observe that the model ('SearchNovo w/o pseudo reference PSM') exhibits significantly poorer performance on spectra that lack a similar reference in the database. Specifically, when the input spectrum is matched to a reference PSM with low similarity (e.g., similarity scores between 0 and 0.3 in Figure 5), the retrieved reference peptide sequence often deviates substantially from the target peptide sequence. This mismatch provides minimal useful information for target peptide sequence generation, leading to a notable decline in the performance.

To further address this issue, we propose a simple-yet-effective method that can prevent the model from over-relying on dissimilar (noisy) reference spectra when generating target peptide sequence. Specifically, for each query PSM $(\mathbf{x}, \mathbf{y})$ in the training dataset, given that its retrieved reference PSM is $(\mathbf{x}_s, \mathbf{y}_s)$, we construct a pseudo reference PSM as $(\texttt{[PAD]}, \texttt{[EOP]})$, where the special token $\texttt{[PAD]}, \texttt{[EOP]}$ create an empty reference PSM and it teaches the model to generate the target peptide sequence without relying on the true reference PSM $(\mathbf{x}_s, \mathbf{y}_s)$. Formally, we minimize the following joint loss function for SearchNovo:

$$\ell(\theta) = -\sum_{j=1}^{N}(\log p(y_j \mid \mathbf{y}_{<j}, \mathbf{x}, \mathbf{x}_s, \mathbf{y}_s; \theta) + \lambda \times \log p\left(y_j \mid \mathbf{y}_{<j}, \mathbf{x}, \texttt{[PAD]}, \texttt{[EOP]}; \theta\right)), \tag{10}$$

where $\lambda > 0$ is a trade-off coefficient. In experiments, we implement the similarity $\mathrm{sim}(\mathbf{x}, (\texttt{[PAD]}, \texttt{[EOP]}))$ in Eq.6 as 1. This makes it possible that a single unified model can handle

both scenarios where reference PSM is similar or dissimilar to the query spectrum. Kindly note that the second term of Eq. 10, i.e. $\log p\left(\mathbf{y}_j \mid \mathbf{y}_{<j}, \mathbf{x}, [\texttt{PAD}], [\texttt{EOP}]; \theta\right)$, cannot be replaced with $\log p\left(\mathbf{y}_j \mid \mathbf{y}_{<j}, \mathbf{x}; \theta\right)$, as the SearchNovo model requires the reference PSMs as inputs. Even if the pseudo reference PSM is empty, it should still be included with placeholders like ([\texttt{PAD}], [\texttt{EOP}]).

In the inference phase, we regard each mass spectrum in the test set as the query and search against the training set. The retrieved reference peptide sequence along with the prefix target sequence are fed into the fusion layer and the decoder at each decoding step. The decoding process concludes upon predicting the [\texttt{EOP}] token or reaching the predefined maximum peptide length 100 amino acids.

## 5 EXPERIMENTS

### 5.1 DATASETS AND METRICS

Previous studies have evaluated model performance on various datasets, with some using different versions of datasets under the same name (see Appendix A.3 for details). Following NovoBench Zhou et al. (2024), we conducted a comprehensive benchmark of the baseline models and SearchNovo using three datasets: Seven-species, Nine-species, and HC-PT. These datasets represent a range of spectrum resolutions and peptide sources, providing a diverse testing ground for performance comparison. The Seven-species dataset includes low-resolution spectra from seven species, following the leave-one-out approach from DeepNovo (Tran et al., 2017), where the model is trained on six species and tested on yeast. The Nine-species dataset, used in studies like DeepNovo (Tran et al., 2017), PointNovo (Qiao et al., 2021), and Casanovo (Yilmaz et al., 2022), contains high-resolution spectra from nine species and incorporates three post-translational modifications (PTMs), enabling a comprehensive evaluation of model performance. Similarly, we also follow the leave-one-out strategy where we train the model on 8 species and evaluate the model on the left yeast dataset. The HC-PT dataset features high-resolution spectra of synthetic tryptic peptides covering all canonical human proteins and isoforms, including peptides from alternative proteases and human leukocyte antigen (HLA) peptides, with labels derived from MaxQuant's high-confidence search results (Tyanova et al., 2016). In each case, we split the training set 90/10 for training and validation. Kindly note that the target peptides in the test sets of the above 3 datasets are not present in the training sets. The database search methods cannot work on these datasets because they cannot identify unseen or novel peptide sequences. More information of these datasets can be found in Appendix A.3.

We evaluate model predictions using precision at both the amino acid and peptide levels, following previous works (Tran et al., 2017; Qiao et al., 2021; Yilmaz et al., 2022). Amino acid-level precision is calculated as $N_{\text{match}}^{aa}/N_{\text{pred}}^{aa}$, where $N_{\text{match}}^{aa}$ represents the number of correctly predicted amino acids with a mass difference of $< 0.1$ Da and correct prefix or suffix mass within $0.5$ Da. Amino acid-level precision and recall is then defined as $N_{\text{match}}^{aa}/N_{\text{pred}}^{aa}$ and $N_{\text{match}}^{aa}/N_{\text{truth}}^{aa}$, where $N_{\text{pred}}^{aa}$ and $N_{\text{truth}}^{aa}$ represent the number of predicted amino acids in predicted peptide sequences and ground truth peptide sequences, respectively. Similarly, PTMs identification precision and recall can be formulated as $N_{\text{match}}^{ptm}/N_{\text{pred}}^{ptm}$ and $N_{\text{match}}^{ptm}/N_{\text{truth}}^{ptm}$, where $N_{\text{match}}^{ptm}$, $N_{\text{pred}}^{ptm}$ and $N_{\text{truth}}^{ptm}$ denote the number of matched PTMs, predicted amino acids with PTMs and PTMs in ground truth peptide sequence, respectively. Peptide-level precision, the primary performance metric, is $N_{\text{match}}^{p}/N_{\text{all}}^{p}$, where a predicted peptide is correct only if all amino acids match the ground truth. Given the peptide recall and precision, we also use the area under the precision-recall curve (AUC) as a summary of *de novo* sequencing accuracy.

### 5.2 BASELINES AND EXPERIMENTAL SETUPS

We use 5 representative models as baselines in our experiments: DeepNovo, PointNovo, InstaNovo, AdaNovo and Casanovo, which we have introduced in the related work section. Although SearchNovo enjoys the advantages of both database search and *de novo* sequencing, it is fundamentally a *de novo* sequencing method. Moreover, database search methods are not applicable to these datasets where the test peptides are not present in the training set. Therefore, we did not include a comparison with database search methods. For training SearchNovo, we used a batch size of 32 and trained the model for 30 epochs on an Nvidia A100 GPU. The learning rate was set to 0.0004 with a linear warm-up schedule, and gradient updates were performed using the Adam optimizer (Kingma & Ba, 2014). Optimal hyperparameters were selected based on the validation set. For the baseline models, we used

the original hyperparameters from their respective papers. DeepNovo and PointNovo were validated every 3,000 steps, while the remaining models were validated every 50,000 steps.

## 5.3 MAIN RESULTS

**SearchNovo outperforms state-of-the-art methods on 3 benchmarking datasets.** As shown in Table 1, SearchNovo shows notable superiority over previous peptide sequencing tools in terms of both amino acid-level and peptide-level metrics. Also, we observe that SearchNovo also outperforms other competitors in PTMs identification in Table 2, probably because that the retrieved reference peptide sequence may contain some PTMs that are difficult to be identified, which provide valuable clues for the PTMs identification. Also, we can observe that the overall performance on high-resolution Nine-species and HC-PT datasets is significantly superior to the performance on low-resolution Seven-species dataset. This phenomenon indicates that higher-resolution data provides more detailed spectral information, which enhances the model's ability to accurately infer the peptide sequences.

Table 1: An empirical comparison of models based on amino acid-level and peptide-level metrics. The top-performing model is highlighted in **bold**, while the runner-up is underlined. We trained five models with different random initializations and reported the standard deviation on the HC-PT dataset. For the other datasets, standard deviations were not reported, as training multiple models on all datasets would be computationally extensive.

| Method | Peptide-level performance | | | | | | Amino acid-level performance | | | | | |
| | Seven-species | | Nine-species | | HC-PT | | Seven-species | | Nine-species | | HC-PT | |
| | Prec. | AUC | Prec. | AUC | Prec. | AUC | Prec. | Recall | Prec. | Recall | Prec. | Recall |
|---|---|---|---|---|---|---|---|---|---|---|---|---|
| DeepNovo | 0.204 | 0.136 | 0.428 | 0.376 | 0.313 ± 0.014 | 0.255 ± 0.010 | **0.492** | 0.433 | 0.696 | 0.638 | 0.531 ± 0.018 | 0.534 ± 0.015 |
| PointNovo | 0.022 | 0.007 | 0.480 | 0.436 | 0.419 ± 0.008 | 0.373 ± 0.011 | 0.196 | 0.169 | 0.740 | 0.671 | 0.623 ± 0.015 | 0.622 ± 0.009 |
| InstaNovo | 0.031 | 0.009 | 0.164 | 0.123 | 0.057 ± 0.008 | 0.034 ± 0.010 | 0.192 | 0.176 | 0.420 | 0.395 | 0.289 ± 0.006 | 0.285 ± 0.009 |
| AdaNovo | 0.174 | 0.135 | 0.505 | 0.469 | 0.212 ± 0.022 | 0.178 ± 0.015 | 0.379 | 0.385 | 0.698 | 0.709 | 0.442 ± 0.017 | 0.451 ± 0.023 |
| Casanovo | 0.119 | 0.084 | 0.481 | 0.439 | 0.211 ± 0.010 | 0.177 ± 0.014 | 0.322 | 0.327 | 0.697 | 0.696 | 0.442 ± 0.016 | 0.453 ± 0.022 |
| SearchNovo | **0.259** | **0.174** | **0.550** | **0.489** | **0.447** ± 0.013 | **0.413** ± 0.010 | 0.489 | **0.488** | **0.748** | **0.746** | **0.652** ± 0.008 | **0.658** ± 0.016 |

Table 2: An empirical comparison of models in terms of their ability to identify PTMs. The best results and the second best are highlighted with **bold** and underline, respectively.

| Method | PTM Recall | | | PTM Prec. | | |
| | Seven-species | Nine-species | HC-PT | Seven-species | Nine-species | HC-PT |
|---|---|---|---|---|---|---|
| DeepNovo | 0.373 | 0.529 | 0.615 ± 0.018 | 0.391 | 0.576 | 0.626 ± 0.013 |
| PointNovo | 0.094 | 0.546 | 0.740 ± 0.009 | 0.117 | 0.629 | 0.676 ± 0.012 |
| InstaNovo | 0.115 | 0.294 | 0.261 ± 0.010 | 0.126 | 0.443 | 0.350 ± 0.012 |
| AdaNovo | 0.321 | 0.570 | 0.482 ± 0.022 | 0.448 | 0.652 | 0.552 ± 0.017 |
| Casanovo | 0.251 | 0.566 | 0.460 ± 0.015 | 0.360 | 0.706 | 0.501 ± 0.017 |
| SearchNovo | **0.447** | **0.599** | **0.772** ± 0.011 | **0.472** | **0.764** | **0.715** ± 0.016 |

## 5.4 WHY SEARCHNOVO CAN ACHIEVE SUPERIOR PERFORMANCE?

Table 3: Comparisons between the target peptide sequence of query spectra and the retrieved reference peptide sequence from the seven species dataset. '(+15.99)' and '(+.98)' denote the oxidation-modified (a kind of PTM) and isotopic labeling amino acids before itself, respectively. The overlapping fragments are underlined.

| Target Peptide Sequence of Query Spectra | Retrieved Reference Peptide Sequence |
|---|---|
| IIDASHR | IIGPGINK |
| AGWQGTVTF | AGWQGTITF |
| LTANDVFRK | LTANDIFRK |
| ATPIAEAMMAIVIIDCIIR | ATPIAEAM(+15.99) MAIVIIDOIIR |
| NGAIIAAVQQEGEEIMIISDQGIIVR | NGAIIAAVQ(+.98)Q(+.98)EGEEIMIISDQGIIVR |

In this subsection, we aim to investigate why SearchNovo can achieve superior performance. Our findings are as follows:

**(i) Many of the retrieved reference peptide sequences and the target sequences share overlapping fragments, providing important clues for generating the target sequence.** We provide some

examplar target sequence of query spectra and their corresponding reference peptide sequence from the seven-species dataset in Table 3. As can be observed, there exists notable overlaps between the pairs and thus the latter provides important clues to guide the generation of the former. Also, their shared fragments demonstrate various degrees of alignment. In some cases, there is a near-perfect match (e.g., LTANDVFRK vs. LTANDIFRK), while in others, more differences are apparent. This phenomenon necessitates the robust methods to prevent the models from over-relying on the noisy (dissimilar) reference PSM when generating the target peptide sequence.

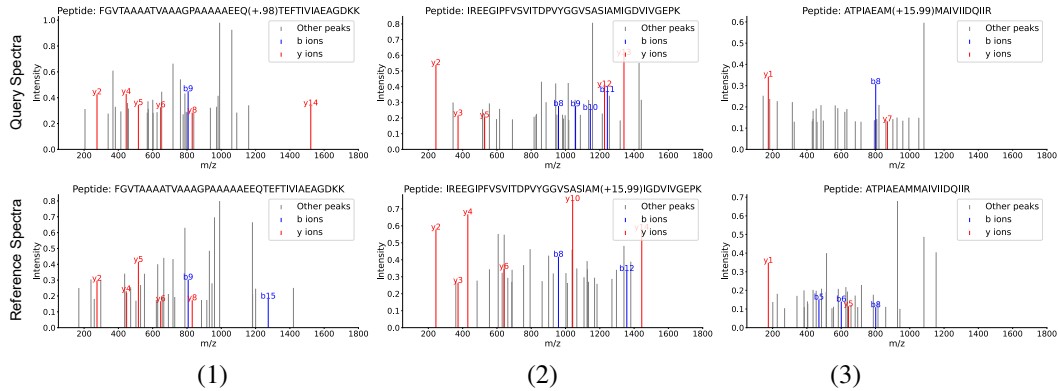

Figure 4: Comparisons between the query spectra and the retrieved reference spectra. The spectra in the same column represent query-reference pairs.

**(ii) The retrieved reference spectra by SearchNovo include some missing signal peaks in the query spectra, enabling SearchNovo to mitigate the issue of missing peaks.** To verify this point, we annotate the b, y ions (signal peaks) using the pyteomics tool (Goloborodko et al., 2013). Specifically, we generate the theoretical b and y ions for the peptide sequence, which represent the expected fragment ions formed from the peptide backbone cleavage during tandem mass spectrometry (MS/MS). Detailed explanations of b and y ions can be seen in Appendix A.5. And then, we compare each peak in the query spectrum to the theoretical ions, and the closest match is identified using a defined tolerance ($\pm$ 1.0 Da in our experiments). If an observed peak matches a theoretical b or y ion within the tolerance, it is classified accordingly. As shown in Figure 4, the retrieved spectra share some common b, y ions (e.g., y2, y4, y5, y6, y8 and b9 in Figure 4(1)) with the corresponding query spectra, indicating that the retriever can identify similar spectra from the database. More importantly, the retrieved reference spectra contain some missing peaks (e.g., y4, y6, y10, b12, and y14 in Figure 4(2)) that should theoretically be present in the query spectra. And thus, they offer valuable hints that aid in inferring the peptide sequences beyond what is available in the query spectra.

## 5.5 ABLATION STUDY

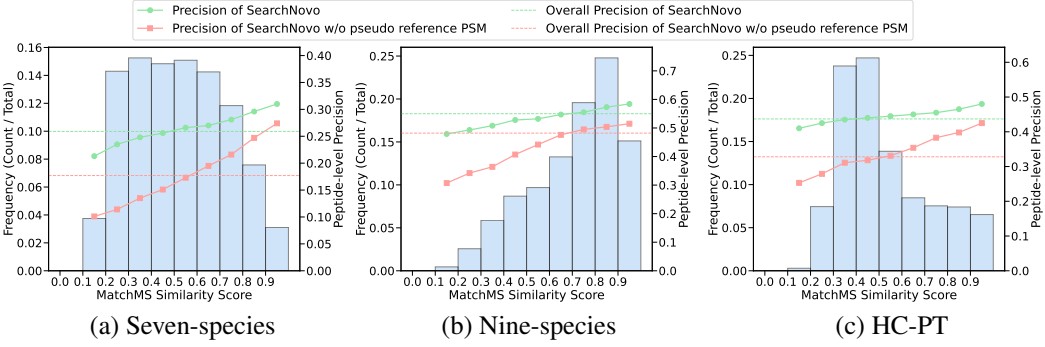

Figure 5: The MatchMS similarity score distribution and the peptide-level precision over different similarity intervals.

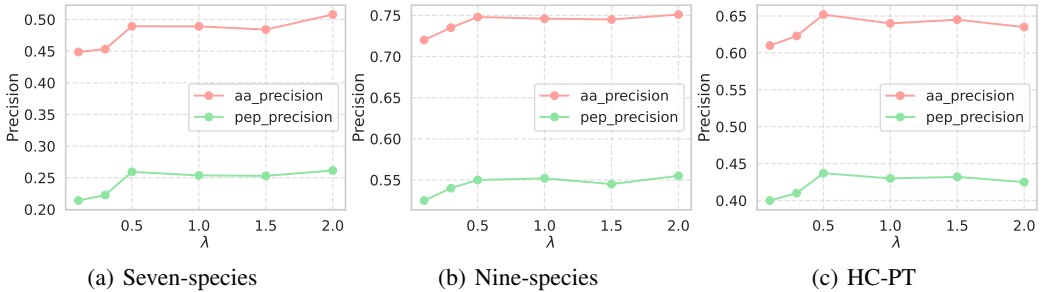

| (a) Seven-species | (b) Nine-species | (c) HC-PT |

Figure 6: Amino acid-level (denoted as aa_precision in the figure) and peptide-level (denoted as pep_precision in the figure) precision as a function of the hyper-parameter $\lambda$.

**(i) The influence of pseudo reference PSM.** As shown in Figure 5, we plotted the distribution histogram of MatchMS similarity scores between each query spectrum and its reference PSM in the test set. The possible range of MatchMS similarity scores, $[0, 1)$, was divided into ten sub-intervals: $[0, 0.1), [0.1, 0.2), \ldots, [0.9, 1.0)$. It can be seen that SearchNovo without the pseudo reference PSM performs significantly worse in the $[0, 0.3)$ range compared to higher similarity intervals, indicating that noisy reference PSMs adversely affect the model's performance. In contrast, SearchNovo shows consistently strong performance across the entire similarity range of $[0, 1)$, demonstrating that the pseudo reference PSMs effectively mitigate the impact of noisy PSMs.

**(ii) The influence of hyper-parameter $\lambda$.** As illustrated in Figure 6, we observe that key metrics in *de novo* peptide sequencing, such as Peptide-level Precision and Amino Acid-level Precision, generally improve with increasing $\lambda$. However, when $\lambda$ becomes sufficiently large, the model's performance stabilizes, suggesting that the influence of the pseudo reference PSM has reached a saturation point. Based on these observations, we set $\lambda = 0.5$ for our experiments across the three datasets for convenience.

Table 4: Ablations on the similarity in SearchNovo.

| Method | Peptide-level Precision | | | Amino Acids-level Precision | | |
|---|---|---|---|---|---|---|
| | Seven-species | Nine-species | HC-PT | Seven-species | Nine-species | HC-PT |
| SearchNovo w/o the similarity | 0.238 | 0.472 | 0.401 | 0.446 | 0.721 | 0.616 |
| SearchNovo | 0.259 | 0.550 | 0.447 | 0.489 | 0.748 | 0.652 |

**(iii) The influence of the similarity** $\text{sim}(\mathbf{x}, (\mathbf{x}_s, \mathbf{y}_s))$**.** In SearchNovo, we incorporate the similarity into the peptide embedding as in Eq. 6. To investigate the influence of the similarity, we remove it from SearchNovo. As shown in Table 4, the results confirm that the similarity score significantly enhances model performance by providing key information from highly similar reference PSMs, which aids in predicting the target peptide sequence. Removing the similarity leads to a marked decline in performance, underscoring its importance in the SearchNovo framework.

## 6 CONCLUSION

In this paper, we introduced SearchNovo, a unified framework that combines the advantages of database search and *de novo* peptide sequencing to improve peptide sequencing. By incorporating an efficient search strategy and fusion module, SearchNovo effectively mitigates missing signal peaks issue, leveraging reference PSMs for enhanced precision. Additionally, the pseudo reference PSMs prevent the models from over-relying on noisy reference, leading to superior performance across multiple benchmarking datasets. Moving forward, future work could focus on improving SearchNovo's robustness in handling low-quality spectra with multiple noisy peaks and exploring its potential applications in more challenging domains like single-cell proteomics and metaproteomics.

## 7 ACKNOWLEDGEMENTS

This work was supported by National Natural Science Foundation of China Project (No. 623B2086, No. U21A20427), National Science and Technology Major Project (No. 2022ZD0115101), Project (No. WU2022A009) from the Center of Synthetic Biology and Integrated Bioengineering of Westlake University, Project (No. WU2023C019) from the Westlake University Industries of the Future Research Funding, CIE-Tencent Doctoral Research Incentive Project. Finally, Jun wants to thank, in particular, the invaluable love and support from Miss Chen.

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

## A  APPENDIX

### A.1  PEAK EMBEDDING METHODS

Following Casanovo (Yilmaz et al., 2022), we treat each peak $(m_i, t_i)$ from the mass spectrum similarly to a "word" in natural language processing. We represent it by encoding its *m/z* value and intensity separately, and then combine these encodings by summing them. For each peak, the *m/z* value is treated as its position, and positional encoding is applied as inspired by (Vaswani et al., 2017), defined as:

$$f_{ij} = \sin\left(\frac{m_i}{\frac{m_{\max}}{m_{\min}}\left(\frac{m_{\min}}{2\pi}\right)^{2j/d}}\right), \quad \text{for } j \le \frac{d}{2}, \tag{11}$$

$$f_{ij} = \cos\left(\frac{m_i}{\frac{m_{\max}}{m_{\min}}\left(\frac{m_{\min}}{2\pi}\right)^{2j/d}}\right), \quad \text{for } j > \frac{d}{2}, \tag{12}$$

where $f_{ij}$ represents the $j$-th component of the embedding for the $i$-th peak, $d$ is the embedding dimension, and $m_{\max}$ and $m_{\min}$ are constants that we set to 10,000 and 0.001, respectively. These positional embeddings ensure that high-resolution *m/z* values are captured. Following the positional encoding approach of the original transformer (Vaswani et al., 2017), these embeddings help the model focus on variations in *m/z* between peaks, which is essential for accurately determining the peptide sequence. The intensity values $t_i$ are encoded via a linear layer $W_g$, projecting them into a $d$-dimensional space, i.e. $g_i = W_g t_i$, where $W_g \in \mathbb{R}^d$ is the linear layer's weight matrix. The final embedding for each peak $(m_i, t_i)$ is obtained by adding the embeddings for intensity and *m/z*, $h_i = g_i + f_i$. Thus, the input to SearchNovo's spectrum encoder consists of embeddings $\mathbf{h} = \{h_i\}_{i=1}^{M}$, where $M$ is the number of peaks in the spectrum. Similarly, the precursor ion $\mathbf{z} = \{(m_{prec}, c_{prec})\}$ is embedded using the same sinusoidal positional encoding for $m_{prec}$, while the precursor charge state $c_{prec}$ is embedded through a PyTorch embedding layer.

### A.2  MULTI-HEAD ATTENTION (MHA)

We present the specific formulation for the multi-head attention mechanism, $\text{MHA}(\cdot, \cdot)$, designed with $H$ attention heads as follows:

$$\begin{aligned} \text{MHA}(q, \mathbf{u}) &= [\text{Att}\left(q, \phi_j(\mathbf{u}), \psi_j(\mathbf{u})\right)]_{j=1}^{H}, \\ \text{Att}(q, \mathbf{u}, \mathbf{v}) &= \text{softmax}\left(\frac{q\mathbf{u}^\top}{\sqrt{d}}\right)\mathbf{v}. \end{aligned} \tag{13}$$

In this formulation, $q$ represents the query vector, while $\mathbf{u}$ is a matrix composed of two dimensions. The notation $[u_j]_{j=1}^{H}$ indicates the concatenation of all individual vectors $u_j$. The functions $\phi_j$ and

$\psi_j$ refer to two distinct linear transformations that project one matrix into another. The term $\frac{1}{\sqrt{d}}$ serves as a scaling factor, where $d$ denotes the dimensionality of the query vector $q$. We recommend consulting the original Transformer paper (Vaswani et al., 2017) for more details.

## A.3 DATASETS

Table 5: The datasets statistics of the tree datasets in SearchNovo.

| Dataset | precusor m/z | precusor charge | Avg. peaks num. | intensity | peptide len. | PTM class | min m/z | max m/z | train/valid/test num. |
|---|---|---|---|---|---|---|---|---|---|
| Seven-species | 719.07 | 2.42 | 466.05 | 956.17 | 15.79 | 3 | 70.17 | 3997.66 | 317,009 / 17,740 / 17,049 |
| Nine-species | 679.68 | 2.47 | 134.91 | 175082.65 | 15.01 | 3 | 53.03 | 35932.63 | 499,402 / 28,572 / 27,142 |
| HC-PT | 635.32 | 2.31 | 184.21 | 143363.17 | 12.53 | 1 | 99.99 | 1999.99 | 213,284 / 25,718 / 26,536 |

Since Peptide-Spectrum Matches (PSMs) data utilized for training and testing are readily accessible through ProteomeXchange (Vizcaíno et al., 2014), researchers can easily download various sections to benchmark their models for *de novo* peptide sequencing. For instance, the performances of DeepNovo (Tran et al., 2017) and PointNovo (Qiao et al., 2021) have been assessed using the seven-species dataset, while InstaNovo (Eloff et al., 2023) conducts its evaluation on datasets curated by the respective authors. Additionally, there exist different versions of these datasets, leading to discrepancies even for models that claim to use the same dataset. As an example, PointNovo and Casanovo operate on different versions of the Nine-species dataset (MassIVE dataset identifiers: MSV000090982, MSV000081382). These inconsistencies in dataset versions complicate the ability to gauge genuine advancements in the field. To address this issue and ensure a fair comparison of models, we re-evaluated the performance of different models comprehensively and consistently across three datasets. Detailed information about these datasets is provided in Table 5.

## A.4 RELATED WORK: RETRIEVAL AUGMENTED GENERATION

SearchNovo is inspired by recent advances in RAG that integrate retrieval mechanisms into various generative Natural Language Processing NLP and vision tasks, such as language modeling(Guu et al., 2020; Borgeaud et al., 2022), image generation Tseng et al. (2020); Sheynin et al. (2022) and Video captioning Whitehead et al. (2018); Xu et al. (2024). In computational biochemistry, retrieval-based strategies are also crucial, such as in multiple sequence alignment (MSA), where relevant protein sequences are retrieved and aligned, playing a fundamental role in methods like MSA Transformer (Rao et al., 2021) and AlphaFold (Jumper et al., 2021). RetMol (Wang et al., 2023) employs retrieved molecules with desired properties to guide the model to realize controllable molecular generation. In contrast to these works, we propose new search (retrieval) strategy tailored for mass spectra data and design novel methods to exploit the reference peptide sequence. SearchNovo integrates database search and *de novo* sequencing into a unified framework, enjoying the advantages of both worlds.

## A.5 EXPLANATIONS OF B, Y IONS.

In mass spectrometry-based proteomics, peptides fragment into ions that provide information about their sequence. Two commonly observed ion types are **b-ions** and **y-ions**. **b-ions** are formed when the peptide bond breaks between the nitrogen and the alpha-carbon (N-C$\alpha$) of the peptide backbone, leaving the charge on the N-terminal fragment. For a peptide sequence, b-ions correspond to fragments starting from the N-terminus. **y-ions**, on the other hand, form when the bond breaks between the carbonyl carbon and nitrogen (C-N), leaving the charge on the C-terminal fragment. These ions represent fragments starting from the C-terminus. By analyzing the pattern of b- and y-ions, peptide sequences can be reconstructed, providing critical insights in *de novo* peptide sequencing and database searches.

For example, consider a peptide sequence `A-G-E-W` (Alanine-Glycine-Glutamic acid-Tryptophan). During fragmentation, the following ions might be observed:

- **b-ions:** `b1 = A, b2 = A-G, b3 = A-G-E`

- **y-ions:** `y1 = W, y2 = E-W, y3 = G-E-W`

In this example, b-ions correspond to fragments from the N-terminus, while y-ions represent fragments from the C-terminus. The complementary information from both ion types allows for the reconstruction of the full peptide sequence.

