# OpenReview forum: "Bridging the Gap between Database Search and \emph{De Novo} Peptide Sequencing with SearchNovo"
_ICLR.cc/2025/Conference — ICLR 2025 Poster_

### Official Review · Reviewer_tavc · 2024-10-22

**Soundness:** 3
**Presentation:** 3
**Contribution:** 3
**Rating:** 6
**Confidence:** 4

**Summary:**

The authors introduce an approach called searchnovo for the identification of peptide sequences of mass spectra. Without doubt, the identification of peptides remains a tough and important question and the authors contribute an approach that allows the incorporation of prior sequence knowledge into de novo sequencing approaches. They compare and benchmark their approach against commonly used de novo approaches on standard benchmarking data sets of the community and show superior performance

**Strengths:**

-  The authors go beyond the thinking of current de novo approaches and allow the inclusion of prior knowledge

-  The authors demonstrate superior performance to common de novo approaches

-  The authors provide potential causal rationalization for the improved performance

**Weaknesses:**

- mixture approaches between denovo and database search have been commonly described (most noteworthy Spider 10.1109/CSB.2004.1332434, BICEPS 10.1074/mcp.M111.014167) also in a formal problem definition, this is by no means an innovation of this paper, but has been around for two decades and should be stated as related work.

- error-tolerant and open database searches are commonly used in the field and also not mentioned here, although (see next point) they may be highly relevant

- the authors argue that database comparison is not valid, at the same time feed prior information from databases into searchnovo that are not available to any other de novo approach. In this way, this is not a fair comparison.

**Questions:**

- Parameters such as the +-20 Da threshold for similar searches appear arbitrarily set. Is there a deeper reason?

- it is not clear to me what database information is provided to searchnovo in the various benchmark. In particular, what is the size of the database and to which extend are related sequences filtered out (are only absolutely identical ones filtered out or also those containing PTMs/chemical modifications/single amino acid substitions)?

- can the authors compare to an error-tolerant/open database search approach (eg msfragger as current de facto gold standard in the field)? It could make sense to replicate a common scenario, e.g. in the 7/9 species setup that I can provide a complete database of the closed related organism. To give an example, human and chicken are evolutionary 300 million years apart, but it has been shown that error-tolerant approaches can overcome that distance, so unless going for very obscure organisms, there are related species available and that should be the starting point.

- for practical purposes in proteomics, reliability is a key issue and the setup of the benchmark is of limited use for proteomics applications. Most mass spectra are of low quality and redundance of measurements is very high (almost all proteins are identified based on multiple spectra).  It is common standard (and even required for publications in practically all venues of the field) that peptide identifications are at least above a 1% false discovery rate threshold. Currently the authors only provide precision/recall metrics, which does not allow judging the number of peptide that the search algorithm itself considers reliable – it is by no means necessary that each spectrum is identified. Can the authors provide in addition a number of peptides above 1% FDR.

---

### Official Review · Reviewer_UgvV · 2024-10-31

**Soundness:** 2
**Presentation:** 2
**Contribution:** 2
**Rating:** 6
**Confidence:** 4

**Summary:**

The authors propose a new method, SearchNovo, for identifying the underlying amino acid sequences of peptides from tandem MS/MS data. This is an important problem, crucial for many applications, such as antigen and antibody discovery. This method attempts to overcome the limitations of de novo methods, which identify peptide sequences from the spectrum alone, and database search methods, which rely on a reference database of spectra to find similar spectra. SearchNovo uses a transformer architecture and concepts from the “retrieval augmented generation” field to seamlessly and organically “fuse” the de novo signal with the signal from similar spectra found in a database.

**Strengths:**

- SearchNovo is an interesting method that combines recent transformer-based advancements in de novo MS/MS-based peptide discovery with new concepts from retrieval-augmented generation, and it has the potential to advance the state of the art in peptide discovery.
- The authors have performed a large assessment study with other methods to demonstrate the contributions of their method.

**Weaknesses:**

**Major concerns**
- My biggest concern is that the performance numbers reported for Casanovo (Yilmaz et al., 2022) deviate significantly from those reported in this paper. For example, focusing on the “Nine-species” dataset, both reported precision and AUPRC differ significantly.
- Comparing de novo methods with hybrid (de novo + database search) methods is challenging, as all homologous sequences must be removed from the training/reference sets. It is not surprising that the reported ablation study (Section 5.3-iii) shows a significant performance drop when the “similarity” of the enumerated PSMs is ignored. A more balanced test set could involve immunopeptidome or metaproteomes.
- I find the underlying mechanics of the proposed model insufficiently explained; a toy example would have helped (e.g., Fig. 1 of Yilmaz et al., 2022, is a good template).
- Equation (2) represents the negative log-likelihood, not cross-entropy.
- Although dataset versions differ as outlined in Section A.3, the drop in peptide-level precision—from 0.81 to 0.481 for Casanovo and from 0.74 to 0.480 for PointNovo (the former metrics reported by Yilmaz et al., 2022)—is concerning.
- The peak encoder (Section A.1) seems heavily inspired by (if not identical to) Yilmaz et al., 2022, and maybe I am missing something, but I don’t see appropriate attribution.
- The authors could improve reliability by using bootstrapping to compute confidence intervals or standard deviations.

**Minor points**
- “de novo” should always appear together without breaking across lines; perhaps \emph{de~novo} would help.
- References in the “Introduction” do not fully represent recent developments in MS/MS data analysis, especially for database search methods.
- In line 160, “PTM” is used before it is defined in line 213.
- In line 188, is the beam search citation correct? “(Science, 1977)”
- In Fig. 3, both “PEPTID” and “PEITED” are misspelled at the top of the figure.
- Some sections (e.g., 5.3 and 5.4) begin with table captions rather than text.
- In Eq. (10), the first “y_j” should be bold.

**Questions:**

Following the points outlined above, it would be helpful if the authors explained *exactly* what differs in their evaluations compared to previous reports on the 'nine-species' dataset. The discrepancies are significant and may impact reviewers' decisions.

---

### Official Review · Reviewer_yzQz · 2024-11-02

**Soundness:** 3
**Presentation:** 4
**Contribution:** 3
**Rating:** 8
**Confidence:** 4

**Summary:**

This paper introduces a new method for predicting peptide sequences from their MS/MS spectra, named SearchNovo. SearchNovo combines two main paradigms in computational proteomics: database search and de novo sequencing. Inspired by the success of retrieval-augmented generation, the authors propose a new transformer-based architecture that predicts a peptide sequence given its MS/MS spectrum and a reference peptide sequence corresponding to the most similar spectrum in a database. The effectiveness of this method is demonstrated empirically on three datasets across multiple metrics. Additionally, the authors discuss why SearchNovo outperforms purely de novo approaches, providing clear qualitative results. The text is well-written and easy to follow. Supplementary materials provide the corresponding source code.

**Strengths:**

Originality

The originality of the work lies primarily in the combination of the de novo and retrieval paradigms for annotating peptide MS/MS spectra within a unified machine learning framework.

Quality

The paper is of very high quality. The authors carefully prepare evaluation datasets, ablate key model components, analyze results in depth, and provide sound justifications for why their model may outperform prior methods.

Clarity

The work is highly clear, with a logical structure that makes it easy to follow from beginning to end.

Significance

This work presents a significant finding by introducing a conceptually new combination of two independent approaches, opening up a new direction for the further research. The advantage of the new method is supported by strong empirical results.

**Weaknesses:**

Two concerns arise from the study:

- Section 5.4 states, “(ii) The retrieved reference spectra by SearchNovo include some missing signal peaks in the query spectra, enabling SearchNovo to mitigate the issue of missing peaks.” and Figure 4 provides the corresponding examples. However, the architecture of SearchNovo does not appear to use the retrieved spectra as input but only the peptide sequence corresponding to the most similar spectrum. It is unclear how the model mitigates the issue of missing peaks without directly incorporating the retrieved spectra.

- Although the authors explain their choice not to include database retrieval baselines, this would be very valuable. Specifically, the authors could show the performance of assigning a sequence to a spectrum based solely on matchms (e.g., using a k-NN classifier with matchms as a metric). As Table 3 suggests, such a model could still achieve high per-amino-acid metrics. Although it may not achieve non-zero per-peptide metrics, it would still provide context on the extent to which the de novo component contributes to the model's performance. Without such results, it is not clear how, for example, the change of the database for retrieval would affect the performance.

**Questions:**

- Figure 5: Why does SearchNovo with pseudo reference PSMs outperform the model without pseudo reference PSMs, even when spectra with high similarity scores (e.g., > 0.8) are available?

- The lambda hyperparameter adds complexity to the model and requires two forward passes. Did the authors experiment with simpler approaches that replace this additional hyperparameter, perhaps by directly using matchms similarity? While the similarity is used in Equation 6, using it, for example, to weight the second term in Equation 8 might provide a more straightforward approach.

- The computational efficiency results in Table 6 provide a valuable complement to the performance metrics presented in the main text. Since the model employs a database search, one could think that it would be significantly slower than the purely de novo methods. I suggest mentioning these results in the main text for added emphasis.

---

### Official Review · Reviewer_xcET · 2024-11-02

**Soundness:** 3
**Presentation:** 3
**Contribution:** 3
**Rating:** 6
**Confidence:** 4

**Summary:**

The paper proposes a novel approach to combine two dominant methods for predicting peptide sequences from tandem mass spectra—database search and machine learning-based *de novo* generation—through a retrieval-augmented generation paradigm.

**Strengths:**

**Originality**. The idea of linking database search with de novo generation is original in the context of peptide sequencing from mass spectrometry data.

**Quality**. The experiments are well-designed, with ablations and qualitative examples that provide valuable insights into the method’s functionality.

**Clarity**. The paper is generally well-written and clear.

**Significance**. The task of peptide sequencing from tandem mass spectrometry data is highly important. This paper demonstrates that merging the two paradigms offers promising new opportunities in this area.

**Weaknesses:**

**Major Comments**

- Quantitative analyses in section 5.4 are missing. The section lacks quantitative experiments to support the qualitative examples regarding why the method works.
- Database search baselines are missing. Implementing at least a simple database search baseline would be beneficial. Even if test set contains distinct peptide sequences, meaning that per-sequence precision is ensured to be 0, per-amino-acid evaluation could provide interesting isights.
- Potential data leakage. Please analyze the overlap between spectra and peptides in the training and test sets. This could be assessed using vector similarity between binned spectra and sequence identity upon alignment to evaluate spectral and peptide similarity, respectively.

**Minor Comments**

- Line 378: Please specify the validation set used.
- Appendix A.1: Please add citations for related works that use similar approaches to encoding mass spectra, such as [1].
- Line 248: Typo: “summarizing AA. Embedding and Pos. Embedding” should be corrected to “summing” and upper-case letter should be fixed.
- Line 291: Same typo as above.
- Equation 7: Comma instead of a dot at the end.
- Line 304: Typo: “can redefined”.
- Equation 10: Dot instead of a comma at the end.

**References**

[1] Yilmaz, Melih, et al. "De novo mass spectrometry peptide sequencing with a transformer model." International Conference on Machine Learning. PMLR, 2022.

**Questions:**

- Could retrieving more than 1 reference sequence be beneficial?

---

### Public Comment · ~COMMENTOKKKK1 · 2024-11-17

The performance of searchnovo is significantly lower than models such as contranovo (https://arxiv.org/pdf/2312.11584) and casanovo v2 (https://www.nature.com/articles/s41467-024-49731-x). The baselines in this paper are all outdated and have inferior performance. In line 408, the statement "outperforms state-of-the-art methods" is not valid. This leads to the inconceivable effectiveness of the proposed method.

The leakage of information from the spectral database can result in an unfair comparison.

Could reviewer yzQz03 further elaborate on more strengths? The strengths listed do not seem to support a rating of 8.

As a biologist, I believe that de novo sequencing only works in cases like novel antigens. In such cases, it is very likely that there is no spectrum in the database. Therefore, unless one can obtain a full database for every possible peptide, I think there should be an experiment showing its performance on peptides without spectra in the database.

---

> ### Author Response · Authors · 2024-11-22
> **Response to  COMMENTOKKKK**
>
> Thanks for your comments. We have provided detailed responses to each point as follows:
>
> >*The performance of SearchNovo is significantly lower than models such as ContraNovo and Casanovo v2.*
>
> 1. The performance differences you mentioned are primarily due to variations in the training and testing datasets. For example, both CasaNovo v2 [1] and ContraNovo [2] use the MassIVE-KB spectral library (identifier: MSV000081142) as their dataset. This dataset is not only completely different from the dataset used by SearchNovo but is also significantly larger, rendering a direct comparison both unreasonable and meaningless. Additionally, the test datasets are also different: our study adopts the widely-used Nine-species dataset from [6] (identifier: MSV000081382). In contrast, the Nine-species dataset utilized by ContraNovo and CasaNovo v2 corresponds to entirely different versions (identifier: MSV000090982).
>
> 2. In ContraNovo, the nine-species dataset has an issue of data leakage, where peptide sequences in the test set may also appear in the training set. Our reported result is lower because, in our version of the nine-species dataset, the training and test sets do not overlap, whereas ContraNovo and Casanovo v2's test set includes peptide sequences from their training set. In other words, our evaluation datasets are more challenging. Therefore, the results we report on the nine-species dataset are lower.
>
> 3. According to ICLR's policy (https://iclr.cc/Conferences/2025/ReviewerGuide), papers published after July 1, 2024, are considered contemporaneous and do not require comparison. Casanovo v2[2], published in Nature Communications on July 30, 2024, falls into this category. Hence, we are not obligated to compare SearchNovo with Casanovo v2.
>
>
> >*The baselines in this paper are all outdated...*
>
> **The baselines we compare against, such as HelixNovo[4] (published at BIB in April 2024) and AdaNovo[5] (published at NeurIPS in October 2024), are NOT "outdated" works**. In fact, they were published later than ContraNovo[1] (published at AAAI in March 2024) you mentioned.
>
> > *The leakage of information from the spectral database can result in an unfair comparison.*
>
> 1. **SearchNovo does not utilize any external database beyond the training set.** The spectral database used in its database search is the training set, whose detailed information can be found in Table 6 (Appendix) of the manuscript. **It ensures that the data used in SearchNovo is consistent with other de novo sequencing baselines, making the comparison fair. **
> 2. Furthermore,**there is no overlap between the peptide sequences in the training and test sets used in our experiments, eliminating the possibility of information leakage.**
>
> >*I think there should be an experiment showing its performance on peptides without spectra in the database.*
>
> Since the training and test sets used in our experiments do not overlap in peptide sequences, there are no spectra in the training set that correspond to the peptides in the test set. **In other words, all the experiments in the manuscript evaluate the model's performance on peptides without spectra in the database (training set)**.
>
>
> ---
> [1] ContraNovo: A Contrastive Learning Approach to Enhance De Novo Peptide Sequencing (Jin et al., AAAI)
>
> [2] Sequence-to-sequence translation from mass spectra to peptides with a transformer model (Yilmaz et al., Nature Communications)
>
> [3] Assembling the community-scale discoverable human proteome. (Wang et al., Cell systems)
>
> [4] Introducing π-HelixNovo for practical large-scale de novo peptide sequencing (Yang et al., BIB)
>
> [5] AdaNovo: Towards Robust De Novo Peptide Sequencing in Proteomics against Data Biases (Xia et al., NeurIPS)
>
> [6] De novo peptide sequencing by deep learning (Tran et al., PNAS)

---

> ### Comment · Reviewer_yzQz · 2024-11-24
>
> As requested by the commenter, I am elaborating on the strengths of the paper, explaining my rating of 8.
>
> In the domain of mass spectrometry, library search and de novo generation are two main paradigms that have been prevalent for at least two decades, yet minimal work has focused on their intersection. This observation applies not only to proteomics but also to metabolomics and related fields. The paper presents a flexible framework that combines these two paradigms, offering a valuable contribution to the field of computational mass spectrometry. Additionally, the paper is clearly written, providing a strong intuition for how this method can be adjusted or applied in new contexts. Comprehensive ablations and analyses further offer interesting insights for future work in this direction.

---

> ### Public Comment · ~COMMENTOKKKK1 · 2024-11-25
>
> What concerns me is the effectiveness of this method. Suppose the performance can truly surpass Casanovo v2 or ContraNovo. That's good. They are all open-source models. You can check their performance. Database search like method can not solve the bottleneck of de novo sequencing for now. Likewise the other paper seems to be submitted by the same team on iclr 2025.

---

> ### Public Comment · ~COMMENTOKKKK1 · 2024-11-26
>
> 1.  Some tricks can result in an improvement on a small dataset but not necessarily on a larger dataset. It's like most people will use ImageNet rather than MNIST nowadays. Improvements in small datasets will NOT help biologists get better results from experiments.
>
> 2.  MassIVE-KB is a useful dataset that contains only human data as I remember. So the problem of leakage of information is negligible on the 9 species dataset except for human and mouse data.
>
> 3. Although HelixNovo and AdaNovo are not published for a long time, the performance is not satisfying as they all use a small dataset(Similar to MNIST).
>
> 4. No leakage of information, that's good.

---

### Meta-Review · Area_Chair_noVw · 2024-12-18

**Metareview:**

This paper proposes SearchNovo, which combines de novo peptide sequencing with database search, where the search outcomes can be used to guide the target sequence generation process, thereby improving the peptide sequencing results.
Overall, reviewers note that the paper is well-written, the proposed idea is well-motivated and interesting, demonstrated to work outperform other existing methods, and shows promising potential.
There were concerns about the fairness of the comparison, potential risk of data leakage, and lack of comparison against baselines.
Most of these major concerns have been addressed by the authors during the discussion period to some extent, but additional investigation and experiments might strengthen the work by addressing any remaining concerns.

**Additional Comments On Reviewer Discussion:**

The authors have actively and extensively addressed the reviewers' concerns, including the need for better comparison against baselines, potential risk of data leakage, and comparison against existing relevant methods.
The additional experimental results and clarifications provided by the authors have alleviated the initial concerns of the reviewers, and the reviewers are generally in agreement regarding the potential merit of the proposed method.

---

### Decision · Program_Chairs · 2025-01-22

Accept (Poster)